# Updates on Pulmonary Neuroendocrine Carcinoids: Progress and Perspectives

**DOI:** 10.3390/jcm14165733

**Published:** 2025-08-13

**Authors:** Anna Scognamiglio, Arianna Zappi, Elisa Andrini, Adriana Di Odoardo, Davide Campana, Anna La Salvia, Giuseppe Lamberti

**Affiliations:** 1Department of Medical and Surgical Sciences (DIMEC), Alma Mater Studiorum—University of Bologna, 40126 Bologna, Italy; anna.scognamiglio4@studio.unibo.it (A.S.); arianna.zappi@studio.unibo.it (A.Z.); elisa.andrini3@unibo.it (E.A.); adriana.diodoardo@studio.unibo.it (A.D.O.); davide.campana@unibo.it (D.C.); 2Oncology Unit, IRCCS AOUBO, 40138 Bologna, Italy; 3National Center for Drug Research and Evaluation, National Institute of Health (ISS), 00161 Rome, Italy

**Keywords:** typical carcinoid, atypical carcinoid, lung neuroendocrine tumor, neuroendocrine neoplasm, NET, targeted therapy, treatment

## Abstract

Neuroendocrine neoplasms (NENs) of the lung are a biologically and clinically diverse group of tumors that includes well-differentiated typical and atypical carcinoids (LNETs), as well as poorly differentiated large-cell neuroendocrine carcinoma and small-cell lung cancer. Despite their relative rarity, the incidence of LNETs is increasing, primarily due to advancements in diagnostic techniques and heightened clinical awareness. While the current World Health Organization (WHO) classification offers a morphological basis for diagnosis and prognosis, particularly for extrapulmonary neuroendocrine neoplasms (ep-NENs), it has limitations in predicting the clinical behavior of pulmonary carcinoids. Recent evidence highlights the inadequacy of traditional criteria in fully capturing the biological complexity and clinical heterogeneity of these tumors. This review explores the evolving landscape of LNETs, focusing on well-differentiated forms and analyzing current classification systems, clinicopathological features, and the emerging role of novel prognostic and predictive biomarkers. Advances in histopathology and molecular profiling have begun to elucidate distinct molecular subsets within carcinoids, offering potential avenues for improved risk stratification and therapeutic decision-making. Although there are limited treatment options for advanced disease, new insights into tumor biology could facilitate the development of personalized therapeutic strategies and pave the way for future innovations in LNET management.

## 1. Introduction

Neuroendocrine neoplasms (NENs), traditionally considered rare malignancies, constitute a heterogeneous group of neoplasms arising from neuroendocrine cells located throughout various anatomical sites. In recent decades, their epidemiology has changed markedly: data from population-based registries show that the incidence of NETs has increased from 1.09 per 100,000 individuals in 1973 to 6.98 per 100,000 in 2012 [1] with a further acceleration between 2013 and 2015 observed across multiple subgroups, including patients with localized disease and tumors of the lung [2]. The estimated prevalence reached 171,321 cases in 2014, reflecting both improved diagnostic capabilities and the typically indolent course of many NETs. Five-year overall survival has also improved over time, including in advanced-stage disease, owing to the development and broader availability of more effective systemic therapies [1]; however, their prevalence is higher, reaching about 35 per 100,000, due to both advancements in diagnostic techniques and increased awareness of NENs [3].

Based on histological differentiation and proliferative activity, NENs are classified into two main categories: well-differentiated neoplasms, referred to as neuroendocrine tumors (NETs), and poorly differentiated forms, referred to as neuroendocrine carcinomas (NECs), according to the WHO classification [4]. This framework applies across anatomical sites but exhibits specific nuances depending on the primary location.

NETs most commonly originate in the gastroenteropancreatic (GEP) tract and the lungs, which together account for the majority of cases. Epidemiological data show that the small intestine is the most frequent site of GEP-NETs, followed by the rectum, pancreas, stomach, and appendix [2,5]. In the lungs, neuroendocrine neoplasms (LNENs) represent approximately 25% of primary lung malignancies and include a wide range of biological behaviors [1]. Less frequently, NETs arise in other organs such as the thymus, genitourinary tract, and skin (Merkel cell carcinoma), contributing to the growing overall incidence and prevalence of NENs globally.

In extrapulmonary NETs (ep-NENs), clinical behavior is generally well predicted by current classifications based on tumor morphology and the Ki-67 proliferation index. However, in lung NENs, although the WHO classification distinguishes well-differentiated tumors, i.e., typical carcinoids (TCs) and atypical carcinoids (ACs), collectively referred to as lung NETs (LNETs), from poorly differentiated forms, i.e., large-cell neuroendocrine carcinoma (LCNEC) and small-cell lung carcinoma (SCLC), this dichotomous approach does not fully account for the marked heterogeneity in clinical outcomes observed within LNETs.

Nonetheless, within LNETs, there is a dramatic difference in clinical behavior, not completely explained by the current classifications, which is thus prognostically inefficient. Recently published studies have investigated clinical and biological factors to stratify LNET outcomes beyond the WHO classification that deepened our understanding of LNET biology, with clinical and therapeutic implications.

This review aims to provide an updated overview of LNETs, focusing on their biological and clinical characteristics, advances in diagnostics, and emerging treatment paradigms.

## 2. Current Standard in Diagnosis of LNETs

The histopathological diagnosis of LNETs is based on identifying characteristic morphological features and confirming the neuroendocrine nature of the tumor through immunohistochemical detection of a specific panel of markers according to the 5th edition of the thoracic WHO criteria (2021) (Table 1).

TCs represent the least aggressive LNEN and are characterized by organized growth and regular morphology. The diagnosis is based on a low mitotic index (<2 mitoses per 2 mm^2^) and the absence of necrosis. Although they represent only 1–2% of all lung tumors, these tumors have an excellent prognosis: over 90% of patients are alive five years after diagnosis, and surgery is often curative [6,7,8].

ACs are also well-differentiated in terms of morphology but exhibit more aggressive features, such as focal necrosis and a higher mitotic count (2–10 mitoses per 2 mm^2^). This increased aggressiveness is reflected in their prognosis: approximately 40–50% of patients develop metastases [6,7,8]. The five-year survival rate ranges from 40% to 75% [9,10].

Immunohistochemical markers that aid LNEN diagnosis include Chromogranin A (CgA), synaptophysin (Syn), cluster of differentiation 56 (CD56), and neuron-specific enolase (NSE). Thyroid transcription factor 1 (TTF-1) is frequently used in the immunohistochemical evaluation of LNETs to support a pulmonary origin, yet its diagnostic utility is limited by low sensitivity and heterogeneous expression across tumor subtypes. Indeed, TTF-1 is positive in ~50% of LNETs and is associated with female sex and absence of necrosis [11]. The correlation between TTF-1 positivity and absence of necrosis potentially reflects a less aggressive tumor biology also given the favorable trend in survival.

However, current classification methods have shown inconsistency also across experts in classifying LNETs as TCs or ACs, mostly because of poor reproducibility of the mitotic count. In GEP-NENs, the Ki67 proliferation index is a key parameter in the WHO classification, with a well-established prognostic and predictive role: it allows stratification into G1, G2, and G3 grades, guides therapeutic decisions, and reflects the tumor’s biological aggressiveness [5]. In contrast, in LNETs, Ki67 is not yet a primary diagnostic criterion due to overlapping values among categories and the lack of universally accepted cutoff thresholds. Ki67 with a cutoff of 3% has been shown to add prognostic information in LNETs and to speed up diagnosis, especially if automated or semiautomated platforms are used [12,13]. Although preliminary, these findings support the hypothesis that Ki-67 may provide independent prognostic information beyond classical morphological classification, especially when integrated with other parameters such as the mitotic count and necrosis. Thus, a unified classification has been proposed to also include LNETs but has not yet been adopted [4]. Recently, a machine learning multi-omics-based approach identified a subcategory of ACs with distinct molecular features and intermediate prognosis between ACs and LCNECs, defined as supra-carcinoids, which are discussed below [9].

Positron emission tomography/computed tomography (PET/CT) plays a crucial role in the diagnostic and prognostic assessment of pulmonary carcinoids. The dual-tracer approach using [^68^Ga]-labeled somatostatin analogues (SSAs) and [^18^F]FDG enables in vivo characterization of tumor biology, reflecting differences in somatostatin receptor density and glucose metabolism between TCs and ACs. TCs generally exhibit high uptake of [^68^Ga]-SSAs due to dense somatostatin receptor expression, whereas ACs, characterized by a higher mitotic index and dedifferentiation, show greater avidity for [^18^F]FDG, indicating increased glycolytic activity [14,15,16,17]. This complementary imaging allows differentiation between TCs and ACs, guides preoperative planning, and supports patient selection for radioligand therapy (RLT) [16,18,19]. Moreover, semiquantitative parameters such as the SUV ratio (SUVr) between [^68^Ga]-SSAs and [^18^F]FDG have shown significant predictive value in discriminating tumor subtypes and correlating with clinical outcomes [14,15,17,20]. Accordingly, current guidelines recommend [^68^Ga]-SSA PET/CT as the imaging modality of choice for initial staging of well-differentiated pulmonary neuroendocrine tumors, while [^18^F]FDG PET/CT is advised in suspected or confirmed ACs, in cases of disease progression, or when [^68^Ga]-SSA imaging is negative [21].

## 3. Genomic Characterization

Recent integrative genomic studies have elucidated the molecular landscape of LNETs, through combined analysis of DNA (whole-genome and whole-exome sequencing) and RNA (transcriptome) data. LNETs exhibit a low somatic mutation burden (mean ~0.4 mutations/Mb), markedly lower than that observed in other lung cancers such as SCLC or lung adenocarcinoma. A key feature of LNETs is the high frequency of mutations in genes involved in chromatin remodeling and epigenetic regulation. Mutations in *MEN1*, a tumor suppressor gene that encodes a scaffold protein essential for histone methylation and transcriptional repression via interaction with MLL/MLL2 and SUV39H1, were found in approximately 13% of cases, often accompanied by loss of heterozygosity [22,23]. Similarly, *ARID1A*, a component of the SWI/SNF chromatin-remodeling complex responsible for nucleosome repositioning and transcriptional control, harbors truncating mutations in about 7% of tumors [24]. Other significantly altered genes include *PSIP1* (a chromatin-associated adaptor of MLL complexes), as well as histone methyltransferases (*SETD1B*, *NSD1*, *EZH1*), demethylases (*KDM4A*, *JMJD1C*) [23], and components of the cohesin complex (*STAG1*, *NIPBL*) [25]. Mutations in SWI/SNF subunits—*SMARCA1*, *SMARCA2*, *SMARCA4*, *ARID2*, and *BCL11A*—are mutually exclusive and collectively affect over 20% of cases, further emphasizing the central role of epigenetic deregulation in LNET pathogenesis [24] (Figure 1).

Canonical driver mutations commonly seen in high-grade neuroendocrine carcinomas of the lung, such as *TP53* and *RB1*, are rare (<5%) in LNETs, supporting the hypothesis that these tumors do not represent precursor lesions of typical SCLC but rather follow distinct tumorigenic trajectories [26]. Transcriptomic profiling via RNA sequencing confirms this molecular divergence, with LNETs clustering separately from SCLC and lung adenocarcinomas [9]. Despite this distinct molecular identity, no clear gene expression-based segregation was observed between typical and atypical carcinoids. However, this finding must be interpreted with caution due to the underrepresentation of atypical carcinoids in the studied cohort (*n* = 9 out of 69 total cases), limiting the statistical power to detect subtype-specific molecular signatures [27].

Although SCLC has long been considered a separate entity from LNETs, recent evidence suggest that a subset of SCLC, which lacks of the typical contemporary loss of TP53 and RB1, referred to as atypical SCLC (aSCLC), might actually derive from LNETs [28]. This subset is predominantly found in never smokers or light smokers, which represent <5% of patients with SCLC in Western populations, some of whom had a previous diagnosis of an LNET. Compared to typical SCLC, aSCLC occurs in younger patients, has a lower KI67 and lower TMB, and has a mutational landscape more similar to that of LNETs (e.g., *MEN1*, *EIF1AX,* and *ARID1A*) [7]. In aSCLC, Rb1 and p53 are functionally inactivated by amplification of *MDM2*, *CCND1*, and *CDK4* caused by chromothripsis, a catastrophic genomic rearrangement process mostly involving chromosomes 3, 11, and 12 [28,29]. These tumors have an intermediate prognosis compared to SCLC and ACs.

Supra-carcinoids exhibit a distinctive landscape of genetic mutations, including mutation of BRCA1-associated protein 1 (*BAP1*) and ataxia–telangiectasia mutated (*ATM*), whose products are involved in DNA damage repair, and TP53, a key tumor suppressor gene, which leads to genomic instability and genomic fragility [9]. Also, supra-carcinoids highly express genes in the NOTCH family, such as NOTCH1, NOTCH2, and NOTCH3 [9]. Compared to GEP-NETs, LNETs share mutations in the *MEN1* gene but lack mutations in *ATRX* or *DAXX*, which are common in pancreatic NETs.

## 4. Improving Outcome Prediction: Emerging Prognostic Biomarkers

The inefficiency of the current classification in prognostically stratifying LNETs, especially within the AC category, has fostered research to identify new biomarkers that could significantly improve diagnostic accuracy and patient stratification while also opening new therapeutic avenues.

Multi-omics approaches, i.e., the use of different biological datasets including genomics, transcriptomics, epigenomics, proteomics, and metabolomics, have been applied to investigate the clinically heterogeneous behavior of LNETs, especially ACs. A machine learning-based multi-omics approach identified a subgroup of ACs, defined as supra-carcinoids, which are morphologically not distinguishable but have a poorer prognosis than other ACs [9]. Supra-carcinoids have increased expression of immune checkpoint genes and increased neutrophil infiltration into the tumor microenvironment, suggesting a potential target for the development of immunotherapy-based treatment strategies.

Orthopedia Homeobox Protein (OTP), a key protein for embryonic development of neuroendocrine structures, is a highly specific marker for LNETs, distinguishing them from GEP-NETs [13]. Also, tumors with low or absent OTP expression tend to exhibit more aggressive behavior, a higher metastatic risk, and poorer survival, making it a useful prognostic factor, especially when combined with Ki67 and the mitotic count [9,13].

CD44, a transmembrane glycoprotein that binds hyaluronic acid, triggers multiple intracellular pathways, including PI3K/AKT/mTOR and Ras-MAPK signaling, therefore regulating different biological processes. CD44 loss of expression is associated with increased risk of metastases and poor prognosis independent of other prognostic factors [9,13] and significantly enhances the predictive value for disease progression in LNETs when combined with OTP and Ki67 (with a 5% cutoff) [10].

Angiopoietin-like protein 3 (ANGPTL3), a liver-derived protein involved in lipid metabolism and angiogenesis [30], has shown differential expression in LNETs compared to other high-grade neuroendocrine lung neoplasms such as LCNEC and SCLC [9,30].

HER4 (encoded by *ERBB4*) is a receptor tyrosine kinase, part of the super-family of epidermal growth factor receptors. It triggers the MAPK and PI3K/AKT pathways and is widely expressed in adult and fetal tissues [9,31]. High ANGPTL3 and ERBB4 expression has been found in cluster B carcinoids, a group of LNETs associated with a worse prognosis compared to other LNETs, with 90% of these tumors displaying levels above 1 Fragment Per Kilobase of transcript per Million mapped reads (FPKM) [9].

A three-tier molecular classification has grouped LNETs into the LC1, LC2, and LC3 subtypes, which exhibit distinct gene expression, mutational profiles, and clinical characteristics [32]. This classification incorporates ASCL1, a transcription factor driving neuroendocrine differentiation [33,34], and S100, a small calcium-binding protein expressed in melanoma and some NETs [35].

The LC1 subtype is characterized by high expression of ASCL1, is predominantly found in the peripheral lung, tends to occur in older patients, and shows a low proliferative index, making it closely comparable to TCs. Given their lower metastatic potential and favorable prognosis, LC1 carcinoids generally require less aggressive clinical management [32].

In contrast, the LC2 subtype presents a more complex and intermediate profile, frequently harboring *MEN1* mutations and showing high expression of S100. These tumors can be found in both peripheral and central lung regions, and their clinical behavior overlaps with ACs. While LC2 tumors have a greater risk of disease progression than LC1 tumors, they do not yet exhibit the full aggressive potential of high-grade neuroendocrine tumors [32].

At the most aggressive end of the spectrum, the LC3 subtype lacks ASCL1 and S100 expression and is enriched with high expression of FOXA3 and HNF1A. These tumors predominantly arise in the central lung, often presenting in younger patients and displaying a high metastatic potential. LC3 tumors do not share the genomic instability and TP53/ATM/BAP1 mutations typical of supra-carcinoids [9], yet their distinct transcriptomic profile and clinical features suggest a possible intermediate phenotype within the lung neuroendocrine spectrum [32]. By distinguishing tumors based on molecular features rather than morphology alone, this classification improves risk stratification and prognosis prediction, allowing for more personalized patient management.

Telomerase reverse transcriptase (TERT) is a catalytic subunit of telomerase that preserves telomere length and enables cellular immortality, whose activation is a known feature in high-grade tumors [36]. TERT expression assessed by RNA sequencing is associated with a more advanced stage and worse survival in LNETs, irrespective of stage and morphology (i.e., TCs vs. ACs) [36]. These results were confirmed in an independent validation cohort where TERT-low patients had a 5-year OS of 91.3% compared to 78.8% for TERT-high patients [36]. Interestingly, TERT expression is regulated by methylation of the CpG site cg11625005, which can be assessed in a more clinically available way compared to RNA sequencing (Figure 2) [36].

Liquid biopsy refers to the molecular characterization of tumors through analyses of molecules from the tumor in biological fluids, most often the blood. It allows non-invasive tumor characterization, also capable of capturing tumor heterogeneity, but its sensitivity is highly dependent on tumor characteristics (shedding tumor) and test sensitivity [37,38]. The NETtest is a liquid biopsy test which analyzes the expression by RNA of a panel of 51 specific genes linked to key biological processes in NETs, such as cell proliferation (e.g., Ki67), neuroendocrine differentiation (e.g., *SSTR5*, *TPH1*, *VMAT1*), and tumor signaling (e.g., *BRAF*, *KRAS*, *RAF1*) [37]. The NETest thus provides a score which has been associated with diagnosis, disease recurrence, progression, and outcomes in different types of NENs, including LNETs [37,39]. Nonetheless, limitations of the NETest include the lack of interlaboratory standardization, costs, and accessibility, which hinder its widespread adoption into clinical practice, as extensive validation in large, independent cohorts to confirm its reproducibility in different patient populations is still required.

Alongside molecular biomarkers, readily available clinical features that affect prognosis may aid in patient risk stratification. In a multicenter retrospective study, factors associated with survival outcomes in resected LNETs were used to design the Rachel score nomogram [40]. The score, which includes sex, age, the presence of lymph node metastases, morphology (TC vs. AC), and the Ki67 index, stratifies patients with LNETs into three distinct risk groups: low risk (0–90 points), intermediate risk (91–130 points), and high risk (>130 points). Patients in the three groups identified by the Rachel score have different overall survival and progression-free survival, therefore aiding clinical management and therapeutic planning in these patients.

## 5. Novel Treatment Strategies and Predictive Markers

The therapeutic landscape for metastatic LNETs remains limited, especially because of poor objective response rates (ORRs). Despite the lack of solid prospective data on LNETs, somatostatin analogues (SSAs) are frequently used in SSTR2-positive (by somatostatin receptor imaging or IHC) LNETs. However, the phase III, double-blind, placebo-controlled SPINET study evaluating lanreotide autogel in patients with LNETs was terminated early due to slow patient accrual, rendering it underpowered to demonstrate a survival benefit [41]. Nevertheless, PFS was numerically longer in the lanreotide arm compared to in the placebo arm (16.6 vs. 13.9 months, respectively), which is, however, shorter than the PFS observed in the CLARINET study of lanreotide in patients with GEP-NETs. Everolimus, an mTOR inhibitor, is one of the few approved treatments in LNETs, thanks to the results of the phase III randomized placebo-controlled RADIANT-2 (*n* = 33 LNETs in the everolimus arm) and RADIANT-4 (*n* = 63 LNETs in the everolimus arm) studies [42,43]. In an exploratory subgroup analysis of LNETS patients from the RADIANT-4 study, the ORR was <2%, and the median progression-free survival was 9.2 months in the everolimus arm [44]. In the phase II LUNA trial, the highest 9-month progression-free survival rate with the combination of pasireotide and everolimus (58.5%) was observed, but again, the outcomes were not transformative [45].

Seeking to overcome the limits of everolimus, including the low ORR, treatment strategies with the alkylating agent temozolomide (TEM) have been investigated. In retrospective series, TEM-based therapies demonstrated a potentially interesting ORR of 14%, a DCR of 57%, and an mPFS of 10 months [46]. The ATLANT study was the first prospective phase II trial which evaluated TEM in combination with the SSA lanreotide autogel in 40 patients with LNETs. Nonetheless, the primary endpoint of the disease control rate (DCR) at 9 months was not met; therefore, the trial was formally negative mainly because of its design [46].

Since the combination of TEM with capecitabine (CAPTEM) is more effective than single-agent TEM in pancreatic NETs [47], the combination has been explored also in LNETs. Retrospective series reported an ORR of 30–39%, a PFS of 13–33 months, and an OS ~68 months [48,49]. Also, CAPTEM was associated with a reduced risk of progression or death compared to TEM (PFS 33.9 vs. 15.5 months, respectively; HR 4.01) [49], suggesting that CAPTEM could be more active than single-agent TEM in LNETs, as observed in pancreatic NETs [48]. Furthermore, there is evidence that the methylation status of the O6-methylguanine–DNA methyltransferase (MGMT) promoter might be a potential predictive biomarker for response to TEM-based chemotherapy in advanced, well-differentiated LNETs [50,51]. The prospective MGMT-NET phase II trial tried to demonstrate the MGMT deficiency status (by methylation or IHC) as a predictive biomarker in 105 patients with advanced, well-differentiated NETs (*n* = 38 LNETs) receiving alkylating agents or oxaliplatin-based chemotherapy. The primary endpoint of a ≥35% increase in the 3-month ORR in patients with MGMT-deficient (dMGMT) tumors treated with TEM, compared to MGMT-proficient (pMGMT) tumors, was not met (ORR: 29.4% vs. 8%, *p* = 0.172), making the study formally negative. However, secondary analyses showed that alkylating agents had numerically higher efficacy in dMGMT-NETs (best ORR 52.9%, mPFS 14.6 months) compared to pMGMT-NETs (best ORR 11.5%, mPFS 11.3 months), so the study offers prospective evidence supporting dMGMT, but further studies are needed [52]. Taken together, the available evidence indicates that CAPTEM is an active regimen in LNETs, although prospective validation is warranted. Moreover, tumors with MGMT promoter methylation, preferably assessed through methylation-specific sequencing, appear to derive greater benefit from temozolomide-based therapies.

TEM, with or without capecitabine, is generally recommended as the preferred first-line chemotherapy in NCCN and ESMO guidelines, with platinum-based therapies suggested as a second-line option [53].

RLT is a treatment option for patients with advanced, well-differentiated, SSTR-positive LNETs who have progressed after SSAs, capecitabine or TEM, everolimus, or a combination of these therapies, currently available in some countries, as recommended in the NCCN and ESMO guidelines [21,54]. Evidence in support of the use of RLT in LNETs comes from retrospective studies or ones which only included a relatively small number of patients with LNETs [55,56,57]. In a phase II study of RLT with ^177^Lu-DOTATATE, the ORR was 59%, the median PFS was 18.5 months, and the median OS was 48.6 months [57]. Different outcomes were observed depending on disease characteristics. Indeed, in 15 patients with TCs, the ORR was 33%, the PFS was 20.1 months, and the OS was 48.6 months, whereas in 19 patients with ACs, no objective response was observed (best response was stable disease in 47% of cases), the median PFS was 15.7 months, and the median OS was 37 months [57]. Furthermore, patients whose LNET was TTF-1-positive had significantly shorter PFS on RLT (7.2 vs. 26.3 months), whereas those whose LNET was 18F-FDG-avid had numerically shorter PFS compared to those with an 18F-FDG-negative LNET (15.3 vs. 26.4 months) [57]. This prospective data suggests that TTF-1 positivity, alongside LNET type and likely 18F-FDG positivity, might serve as a prognostic factor in patients with LNETs receiving RLT.

NETs are highly vascular tumors, and frequently mutated genes in NETs are involved in angiogenesis [58]. Thus, several tyrosine kinase inhibitors with antiangiogenic effects, like cabozantinib and axtinib, have been investigated in NETs, including LNETs. Cabozantinib is an inhibitor of multiple tyrosine kinases with a pivotal role in NET pathogenesis, including c-MET and Vascular Endothelial Growth Factor Receptor 2 (VEGFR2). The recently published phase III randomized placebo-controlled CABINET trial included 39 patients in the extrapancreatic NET cohort who had previously progressed on systemic therapy [59]. In this cohort, overall, cabozantinib prolonged the median PFS which was the primary endpoint (8.4 vs. 3.9 months), and subgroup analysis confirmed cabozantinib efficacy also in patients with LNETs (HR 0.17). Nonetheless, the ORR was only 5% on cabozantinib, and the tolerability profile led to 66% of patients requiring dose reductions and 31% of patients discontinuing treatment. The activity of cabozantinib with lanreotide autogel was also investigated in the Italian, multicenter, open-label, phase 2 LOLA study [60]. In this study, the cabozantinib dose was 40 mg daily (compared to 60 mg in the CABINET trial), which improved drug tolerability. The preliminary results reported two partial responses, four stable diseases, and four disease progressions in the LNET cohort. Enrollment has been completed, and the final results are awaited to confirm safety and clinical efficacy.

Pazopanib is a multitargeted agent against VEGFR-1, -2, and -3, platelet-derived growth factor receptor (PDGFR), and proto-oncogene c-Kit. A single-arm phase II study by the Spanish Task Force Group for Neuroendocrine Tumors (GETNE) has evaluated pazopanib after PD of at least one systemic treatment in advanced NETs, which included only five (11.5%) LNETs, making the results of this study poorly applicable to LNETs [61].

Axitinib is a potent and selective VEGFR-1, -2, and -3 inhibitor which plays a central role in angiogenesis. The AXINET study (GETNE1107) is a phase II/III double-blind randomized placebo-controlled trial of axitinib in association with octreotide LAR in patients with advanced G1-2 extrapancreatic NETs, including 28% of patients with LNETs [62]. The preliminary available data have shown a median PFS of 16.6 vs. 9.9 months and an ORR of 13.2% vs. 3.2% in the axitinib vs. placebo arms, respectively. Interestingly, a score based on a nine-gene signature (*RPS10-NUDT3*, *MX2*, *C18orf25*, *SPP1*, *CLDN14*, *REXO1L2P*, *KLHDC3*, *ATXN7*, *CUBN*) was associated with PFS and tumor shrinkage in the axitinib arm but not in the placebo arm [63]. Furthermore, more intense angiogenesis has been observed in right-sided LNETs compared to left-sided ones, suggesting a potential differential activity of antiangiogenic treatments in LNETs, still to be explored [11]. This data supports the potential for developing predictive biomarkers for antiangiogenic therapies in LNETs.

Immunotherapy with immune checkpoint inhibitors (ICIs) targeting cytotoxic T-lymphocyte-associated protein 4 (CTLA-4), programmed cell death protein 1 (PD-1), or its ligand (PD-L1) has yielded suboptimal results in pulmonary carcinoids. This limited efficacy is largely attributable to intrinsic tumor-related factors, including a low tumor mutational burden (TMB), the absence of biomarker-driven patient selection, and the presence of an immunologically “cold” tumor microenvironment.

The tumor microenvironment of LNETs exhibits distinct immunological features that critically shape their immunogenicity and therapeutic responsiveness. One of the most notable characteristics is the limited or absent expression of PD-L1. In certain studies, PD-L1 expression was detected in less than 1% of tumor and T cells [64], although other reports have shown relatively higher PD-L1 levels in LNETs compared to adjacent normal tissue [65]. In addition, lymphocytic infiltration remains sparse in pulmonary carcinoids. Tumor-infiltrating lymphocytes (TILs) are observed in only approximately one-third of cases, with a density ranging from 1% to 10% of cells per high-power field [64]. This paucity of T cells reinforces the concept of an immune “cold” phenotype and is consistent with the absence of mismatch repair protein (MMRp) deficiency, a feature that in other malignancies correlates with heightened responsiveness to PD-1 blockade.

Beyond PD-L1 and TIL status, several metabolic and stromal factors contribute to immune evasion. Notably, the immunosuppressive enzymes indoleamine 2,3-dioxygenase (IDO) and tryptophan 2,3-dioxygenase (TDO) play a key role by catalyzing the degradation of tryptophan into kynurenines, leading to a nutrient-deprived microenvironment that impairs effector T-cell function and promotes the expansion of regulatory T cells (Tregs). IDO expression has been more frequently detected in serotonin-producing NETs, while TDO expression is predominantly confined to the stromal compartment of such tumors [66].

The stromal cells expressing TDO have been identified as cancer-associated fibroblasts (CAFs), which are characterized by strong α-smooth muscle actin (α-SMA) and desmin expression and are located in close proximity to tumor cells. CAFs can inhibit T-cell infiltration and release immunosuppressive cytokines such as transforming growth factor-beta (TGF-β), contributing to immune exclusion [64].

A single-arm phase II trial of single-agent spartalizumab (PDR001), an anti-PD-1 inhibitor, included a cohort of 30 pre-treated patients with LNETs [67,68]. The primary endpoint of the ORR was not met (overall ORR: 7.4%), though in the LNET cohort, it was 16.7%, higher than that of the other NEN cohorts. Thus, the authors concluded that the result in the LNET cohort warrants further investigation. Notably, the ORR was higher in tumors with positive PD-L1 expression on immune or tumor cells (cutoff ≥ 1%), suggesting a potential role as a predictive biomarker in this setting. On the other hand, the NET-002 trial of single-agent avelumab, a PD-L1 inhibitor, included five patients with AC, but no response was observed [69]. The combination of two ICIs, namely ipilimumab, an anti-CTLA-4 monoclonal antibody, and nivolumab, a PD-1 inhibitor, was tested in two trials including patients with LNETs but with opposite results [70,71]. In the nonpancreatic NET cohort of the DART SWOG 1609, 6 patients with an NEN of lung origin were included, but no response was observed among the 14 patients with grade 1 or grade 2 tumors [70]. On the other hand, in a subgroup analysis from the phase II CA209-538 study, the ORR in patients with LNETs was 33%.

The results of combinations of ICIs with other treatment strategies have also been explored. In a phase II trial, the combination of TEM and nivolumab yielded an ORR of 64% in cases of LNETs [72]. In addition, a phase I study exploring the dose escalation of PRRT in combination with nivolumab included two patients with AC, whose best radiological response was stable disease [73].

## 6. Ongoing Clinical Trials

Given the limited treatment landscape and lack of robust clinical guidelines for LNETs, ongoing trials are investigating novel combination strategies to enhance therapeutic efficacy. These include pairing SSAs or chemotherapy, such as TEM, with cabozantinib. The rationale lies in the convergence of their mechanisms on key oncogenic pathways (MAPK, PI3K/Akt/mTOR), angiogenesis, and immune modulation. Cabozantinib may also normalize tumor vasculature, improving drug delivery and overcoming resistance [74,75,76]. These synergistic interactions provide a strong preclinical basis for current clinical studies aimed at expanding treatment options for patients with advanced LNETs.

Indeed, the phase II CABOTEM trial is evaluating the activity and safety of cabozantinib in combination with TEM in patients with LNETs and GEP-NETs progressing after everolimus, sunitinib, or RLT (NCT04893785). Ongoing studies investigating cabozantinib-based combinations that include patients with LNETs are summarized in Table 2.

SSTR expression represents a valuable therapeutic target, and several trials are assessing the efficacy of RLT using both β-emitting isotopes (e.g., ^177^Lu) and α-emitters (e.g., ^212^Pb). Alpha-emitting radionuclides such as ^225^Ac-DOTATATE and ^212^Pb-DOTAMTATE are under investigation for their higher cytotoxic potential and enhanced tumor selectivity compared to conventional beta-emitters. In parallel, novel somatostatin receptor antagonists like ^117^Lu–satoreotide tetraxetan are being studied to improve tumor uptake, particularly in patients with low SSTR density. Efforts to extend the half-life of radiolabeled compounds, such as Evans blue conjugation, are also ongoing, although toxicity remains a limiting concern (Table 3).

Additionally, innovative approaches targeting SSTR include non-peptide drug conjugates such as the first-in-class CRN09682. Preclinical data were presented at the 2024 NANETS meeting, and a global phase I/II study, which will also enroll patients with SSTR-expressing LNETs, is expected to begin soon [77].

## 7. Conclusions

Over the past decade, significant progress has been made in the understanding of LNETs, thanks to advances in histopathology, molecular biology, and clinical research. The identification of novel prognostic and predictive biomarkers is contributing to refining patient stratification and opening new perspectives for a more tailored therapeutic approach. Treatment options remain limited, and leveraging new and existing biomarkers could pave the way to new treatment strategies.

## Figures and Tables

**Figure 1 jcm-14-05733-f001:**
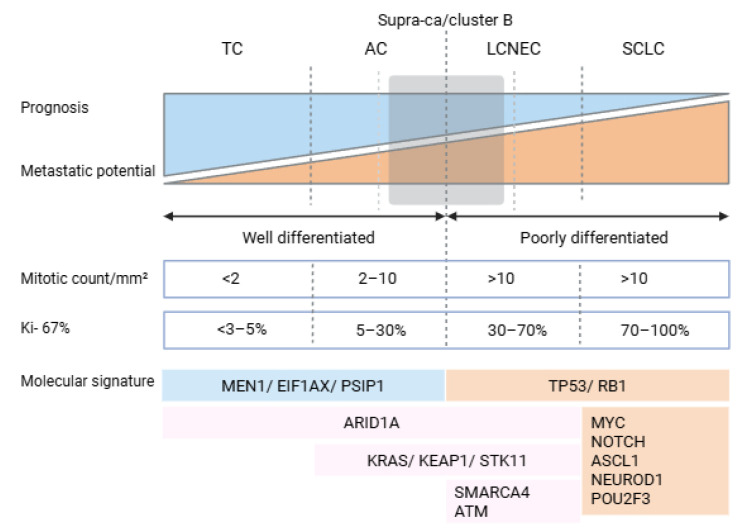
Overview of the most relevant molecular and histological characteristics of lung neuroendocrine neoplasms (LNENs). Abbreviations: MEN1, multiple endocrine neoplasia type 1 gene; EIF1AX, eukaryotic translation initiation factor 1A, X-linked gene; PSIP1, PC4 and SFRS1 interacting protein 1 gene; TP53, tumor protein p53 gene; RB1, retinoblastoma 1 gene; ARID1A, AT-rich interactive domain-containing protein 1A gene; KRAS, KRAS proto-oncogene, GTPase gene; KEAP1, kelch-like ECH-associated protein 1 gene; STK11, serine/threonine kinase 11 gene; SMARCA4, SWI/SNF-related, matrix-associated, actin-dependent regulator of chromatin, subfamily A, member 4 gene; ATM, ATM serine/threonine kinase gene; MYC, MYC proto-oncogene, bHLH transcription factor gene; NOTCH, Notch receptor family gene (typically referring to NOTCH1–4); ASCL1, achaete-scute family bHLH transcription factor 1 gene; NEUROD1, neurogenic differentiation 1 gene; and POU2F3, POU class 2 homeobox 3 gene.

**Figure 2 jcm-14-05733-f002:**
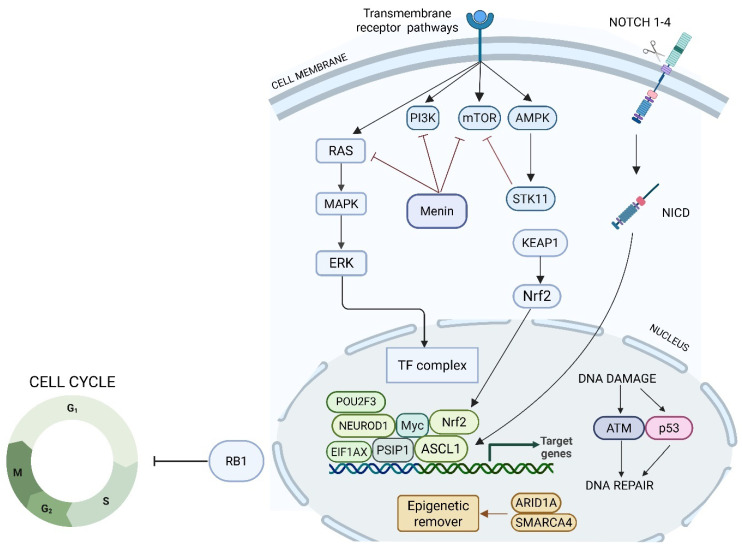
Schema of the interactions among major signaling pathways regulating the tumor cell cycle and DNA repair mechanisms. It highlights the roles of transmembrane receptors, kinases, and transcription factors, with particular focus on the RAS, PI3K, mTOR, AMPK, and Notch pathways. The diagram also shows how these signals converge on the Nrf2 axis, modulating the expression of genes involved in the cellular response to DNA damage, oxidative stress, and repair. The tumor suppressor Rb1 is depicted as a critical checkpoint regulator, controlling cell cycle entry and progression. ARID1A and SMARCA4 regulate chromatin accessibility and transcription of genes involved in DNA repair, cell cycle control, and tumor suppression; their loss is associated with impaired DNA damage response, altered transcriptional programs, and increased tumor aggressiveness.

**Table 1 jcm-14-05733-t001:** The World Health Organization (WHO) 2021 classification of LNENs.

	Grade	Definitional Criteria	Mitotic Index (mitoses/2 mm^2^)
TC	Low grade (G1)	Well-differentiated neuroendocrine morphology, no necrosis	<2
AC	Intermediate grade (G2)	Well-differentiated neuroendocrine morphology, punctate necrosis	2–10
LCNEC	High grade (G3)	Poorly differentiated carcinoma with neuroendocrine morphology, NSCLC cytology (prominent nucleoli and/or moderate to abundant cytoplasm)	>10
SCLC	High grade (G3)	Poorly differentiated carcinoma with neuroendocrine morphology	>10

TC, typical carcinoid; AC, atypical carcinoid; LCNEC, large-cell neuroendocrine carcinoma; SCLC, small-cell lung cancer; mitotic index, number of mitoses per 2 mm^2^.

**Table 2 jcm-14-05733-t002:** Cabozantinib trials.

Trial	Phase	Treatment	NET Type
NCT04427787	II	Cabozantinib plus lanreotide	Well-differentiated gastroenteropancreatic and thoracic NETs
NCT04893785	II	Cabozantinib plus temozolomide	Well-differentiated gastroenteropancreatic and lung neuroendocrine neoplasms
2020–001898–78	II	Cabozantinib plus temozolomide	Well-differentiated gastroenteropancreatic and thoracic neuroendocrine neoplasms

**Table 3 jcm-14-05733-t003:** Peptide Receptor Radiotherapy (PRRT).

Trial	Phase	Treatment	NET Type
NCT05918302	III	[^117^Lu]edotreotide vs. everolimus	NETs, lung neuroendocrine neoplasms, and thymus neoplasms
NCT04665739	II	[^117^Lu]DOTATATE vs. everolimus	Advanced well-differentiated bronchial NETs
NCT05636618	I/IIa	^[212^-Pb] VMT Alpha-Particle Therapy	Advanced SSTR2-positive neuroendocrine tumors
NCT05557708	I	[^212^Pb]pentixather	Atypical and typical carcinoid tumors of the lung and SCLC

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
