# Peer review of "Updates on Pulmonary Neuroendocrine Carcinoids: Progress and Perspectives"

_jcm, 2025, doi:10.3390/jcm14165733_

Round 1

Reviewer 1 Report

Comments and Suggestions for Authors

The authors summarized the current knowledge on pulmonary neuroendocrine tumors (NETs). However, there are several issues that should be addressed:

1. Although the title suggests the focus is on NETs, the manuscript discusses high-grade neuroendocrine neoplasms (such as neuroendocrine carcinomas) extensively. This dilutes the main theme of the review. It is recommended that the authors streamline the content by removing unnecessary sections on high-grade neuroendocrine neoplasms and maintain a clear focus on well-differentiated NETs.

2. Table 1 showed the diagnostic criteria for all pulmonary neuroendocrine neoplasms, not specifically for lung NETs (LNETs), which is not consistent with the table title.

3. While mitotic count is currently the only quantitative criterion for LNET diagnosis, recent studies have investigated the significance of the Ki-67 proliferation index, drawing parallels from its established role in gastrointestinal neuroendocrine neoplasms. It is suggested that the authors make some discussions on these findings.

4. The review compares LNETs with high-grade pulmonary neuroendocrine carcinomas from several aspects. To provide a more comprehensive understanding, it would be valuable to include comparisons between LNETs and NETs from other anatomical sites, such as the gastrointestinal tract.

5. To enhance the readers’ understanding of the molecular differences among neuroendocrine neoplasms of varying grades, it is recommended that the authors make a schematic illustration summarizing the distinct intracellular signaling pathways involved.

6. Since immunotherapies become increasingly important in tumor treatments, which is closely related to tumor microenvironment, the authors are encouraged to expand the discussion on the microenvironment of LNETs, including its implications for therapeutic strategies.

Author Response

The authors summarized the current knowledge on pulmonary neuroendocrine tumors (NETs). However, there are several issues that should be addressed:

Comment 1. Although the title suggests the focus is on NETs, the manuscript discusses high-grade neuroendocrine neoplasms (such as neuroendocrine carcinomas) extensively. This dilutes the main theme of the review. It is recommended that the authors streamline the content by removing unnecessary sections on high-grade neuroendocrine neoplasms and maintain a clear focus on well-differentiated NETs.

Response 1. We thank the reviewer for this suggest. We acknowledge that mentioning LNEC can alter the streamline of the review which is indeed focused on LNETs. Nonetheless, recent evidence reported in the manuscript point towards LNEN being a spectrum, rather than completely different and independent entities. As such, we removed SCLC and LCNEC from the “Current standard in diagnosis of LNETs“ which we agree could be unnecessary, but kept other references to LNECs which we believed were crucial to show commonalities and differences between LNETs and LNECs.

Comment 2. Table 1 showed the diagnostic criteria for all pulmonary neuroendocrine neoplasms, not specifically for lung NETs (LNETs), which is not consistent with the table title.

Response 2. We thank the reviewer for this comment. The title of Table 1 has been modified to reflect the table content.

Comment 3. While mitotic count is currently the only quantitative criterion for LNET diagnosis, recent studies have investigated the significance of the Ki-67 proliferation index, drawing parallels from its established role in gastrointestinal neuroendocrine neoplasms. It is suggested that the authors make some discussions on these findings.

Response 3. We thank the reviewer for this insightful comment, with which we completely agree. We expanded on this concept, which was only mentioned, in the section “Current standard in diagnosis of LNETs”

Comment 4. The review compares LNETs with high-grade pulmonary neuroendocrine carcinomas from several aspects. To provide a more comprehensive understanding, it would be valuable to include comparisons between LNETs and NETs from other anatomical sites, such as the gastrointestinal tract.

Response 4. We thank the reviewer for this suggestion. We added in the manuscript mentions of similarities and differences about biological and clinical differences between LNETs and GEP-NETs.

Comment 5. To enhance the readers’ understanding of the molecular differences among neuroendocrine neoplasms of varying grades, it is recommended that the authors make a schematic illustration summarizing the distinct intracellular signaling pathways involved.

Response 5. We thank the reviewer for this suggestion. We added Figure 2 to the manuscript which is a schema of most relevant pathways altered in LNENs.

Comment 6. Since immunotherapies become increasingly important in tumor treatments, which is closely related to tumor microenvironment, the authors are encouraged to expand the discussion on the microenvironment of LNETs, including its implications for therapeutic strategies

Response 6. We thank the reviewer for this suggestion. In the “Novel treatment strategies and predictive markers” section, LNET tumor microenvironment has been discussed to acknowledge for the scarce response to immune checkpoint-based immunotherapies in LNETs.

Reviewer 2 Report

Comments and Suggestions for Authors

Scognamiglio et. al. summarize existing literature on large neuroendothelial tumors (LNETs), a subset of neuroendocrine neoplasms, describing the limitations of current diagnostic standards and outlining biomarkers, genomic characteristics, and treatments that could help treat LNETS. 

This review manuscript goes into detail and explores the field of LNETs well. The paper could be a useful resource in future diagnostic methods of LNETS, as well as direction for future research into treatment and its identification. The authors describe the differences between types of LNETs distinctly and provide many possible biomarkers and drugs that could be important in furthering diagnosis and therapeutics of LNETs. The paper can be published, with or without the minor concern below.

  1. The Authors in the introduction outline the types of neuroendocrin neoplasms. However, the phrasing of the sentence beginning with “On the other hand, large neuroendocrine neoplasms…” is clunky and it remains hard to tell what items are a subset of what other items given how the sentence is written. A change would make comprehension of the piece easier to digest.

Author Response

Comment 1. Scognamiglio et. al. summarize existing literature on large neuroendothelial tumors (LNETs), a subset of neuroendocrine neoplasms, describing the limitations of current diagnostic standards and outlining biomarkers, genomic characteristics, and treatments that could help treat LNETS. 

This review manuscript goes into detail and explores the field of LNETs well. The paper could be a useful resource in future diagnostic methods of LNETS, as well as direction for future research into treatment and its identification. The authors describe the differences between types of LNETs distinctly and provide many possible biomarkers and drugs that could be important in furthering diagnosis and therapeutics of LNETs. The paper can be published, with or without the minor concern below.

  1. The Authors in the introduction outline the types of neuroendocrin neoplasms. However, the phrasing of the sentence beginning with “On the other hand, large neuroendocrine neoplasms…” is clunky and it remains hard to tell what items are a subset of what other items given how the sentence is written. A change would make comprehension of the piece easier to digest.

Response 1. 

We thank the reviewer for the comments and the suggestion which we addressed by rephrasing the sentence to make the intended meaning clearer to the reader.

Round 2

Reviewer 1 Report

Comments and Suggestions for Authors

The authors have responsed to the questions well and make significant supplements. 

Author Response

Comment: The authors have responsed to the questions well and make significant supplements. 

Response: We sincerely thank the reviewer for their positive feedback and thoughtful evaluation of our revisions.
